# Transcriptomic Analysis of the Host Response to Mild and Severe CTV Strains in Naturally Infected *Citrus sinensis* Orchards

**DOI:** 10.3390/ijms23052435

**Published:** 2022-02-23

**Authors:** José Abrahán Ramírez-Pool, Beatriz Xoconostle-Cázares, Berenice Calderón-Pérez, Enrique Ibarra-Laclette, Emanuel Villafán, Rosalía Lira-Carmona, Roberto Ruiz-Medrano

**Affiliations:** 1Departamento de Biotecnología y Bioingeniería, Centro de Investigación y de Estudios Avanzados del Instituto Politécnico Nacional, Av. IPN 2508, San Pedro Zacatenco 07360, Mexico; jramirezp@cinvestav.mx (J.A.R.-P.); bxoconos@cinvestav.mx (B.X.-C.); bcalderon@cinvestav.mx (B.C.-P.); 2Red de Estudios Moleculares Avanzados, Instituto de Ecología A.C., Carretera Antigua a Coatepec 351, El Haya, Xalapa 91070, Mexico; enrique.ibarra@inecol.mx (E.I.-L.); emanuel.villafan@inecol.mx (E.V.); 3Laboratorio de Virología, UIMEIP, Hospital de Pediatría, Centro Médico Nacional Siglo XXI, IMSS, Av. Cuauhtémoc 330, Col. Doctores, Alcaldía Cuauhtémoc 06720, Mexico; rolica36@yahoo.com

**Keywords:** *Citrus sinensis*, *Citrus tristeza virus*, mild/severe strain, RNA-Seq, differential gene expression

## Abstract

*Citrus tristeza virus* (CTV) is an important threat to the global citrus industry, causing severe economic losses worldwide. The disease management strategies are focused on vector control, tree culling, and the use of resistant varieties and rootstocks. Sweet orange (*Citrus sinensis*) trees showing either severe or mild CTV symptoms have been observed in orchards in Veracruz, Mexico, and were probably caused by different virus strains. To understand these symptomatic differences, transcriptomic analyses were conducted using asymptomatic trees. CTV was confirmed to be associated with infected plants, and mild and severe strains were successfully identified by a polymorphism in the coat protein (CP) encoding gene. RNA-Seq analysis revealed more than 900 significantly differentially expressed genes in response to mild and severe strains, with some overlapping genes. Importantly, multiple sequence reads corresponding to *Citrus exocortis viroid* and *Hop stunt viroid* were found in severe symptomatic and asymptomatic trees, but not in plants with mild symptoms. The differential gene expression profiling obtained in this work provides an overview of molecular behavior in naturally CTV-infected trees. This work may contribute to our understanding of citrus–virus interaction in more natural settings, which can help develop strategies for integrated crop management.

## 1. Introduction

Citrus plants are grown in tropical and subtropical regions of the world, and their fruits are one of the most important crops worldwide. Mexico is one of the leading citrus-producing countries worldwide, and ranks fifth in sweet orange and first in Mexican lime production [1]. However, their production is hampered by detrimental biotic and abiotic factors. Among pathogens that threaten the citrus industry, *Citrus tristeza virus* (CTV) is considered one of the most economically important plant viruses. CTV belongs to the genus *Closterovirus*, family *Closteroviridae*, and boasts one of the largest plant virus genomes (~19.3 kb). This positive-strand RNA virus, which is transmitted by a few aphid species, harbors several open reading frames (ORFs) coding for the replicase complex, a homolog of the cellular heat shock protein of 70 kDa (HSP70) required for cell-to-cell transport and virion assembly, two coat proteins (CP), and other nonessential genes. Some structural and nonstructural proteins, such as CP, p20, and p23, function as suppressors of the plant RNA silencing defense machinery, and are coded close to the genomic RNA 3′ untranslated region (3′ UTR) [2,3]. Phylogenetic analysis of the complete genomic RNA sequences of nine CTV isolates disclosed three clusters that differed in the induced symptoms on the hosts, from mild to severe. In the case of mild isolates, the only symptom observed was yellowing, which did not affect the yield substantially. On the other hand, severe isolates caused stem pitting and dieback of the host, causing important losses of citrus yield [4].

CTV is widely distributed throughout the world and has been said to have transformed the citrus industry, prompting the need for the development and wide distribution of resistant or tolerant rootstocks to control viral infection [4]. Sour orange (*Citrus aurantium*) has been used as a rootstock, given its potential tolerance to CTV infection, replacing *Citrus volkameriana* in Mexico [5,6,7]. However, *C. aurantium* is minimally susceptible to CTV, suggesting partial resistance to this virus. The symptoms range from seedling yellow syndrome to stunting, short internodes, and yellow small-sized leaves caused by different CTV isolates [8,9].

Pathogen infection results in the activation of plant regulators that induce defense mechanisms. The activation of local responses promote long-distance defense signals, which can move to distal tissues [10]. In general, biotrophic pathogens (such as most viruses) induce salicylic acid (SA)-mediated responses followed by an oxidative burst via the activation of a nicotinamide adenine dinucleotide phosphate (NADPH) oxidase. The aforementioned burst induces a cadre of responses which, in incompatible interactions, leads to the death of infected cells. A hallmark of the SA-mediated response is the activation of pathogenesis-related (PR) genes, in particular PR1, which is mediated by the non-expressor of PR genes 1 (NPR1), and downstream by the TGA and WRKY transcriptional activators [11]. Thus, their induction can be used as an indicator of efficient systemic response.

Viroids are small, single-stranded, non-coding, circular RNA particles capable of autonomous replication in plants. Some viroids are important pathogenic agents, whereas others replicate without causing symptoms [3]. Citrus species can be infected by one viroid or multiple pathogens simultaneously, including viruses and viroids. However, only *Citrus exocortis viroid* (CEVd) and *Hop stunt viroid* (HSVd) cause significant losses in citriculture [12]. There are scant studies of mixed viroid and virus/viroid infections currently available.

The understanding of pathogen infection processes is crucial to discern host-pathogen interactions. Transcriptomic analysis, such as gene expression profiling by RNA sequencing (RNA-Seq), provides useful information to better understand host–pathogen interaction. Resistance-related genes and putative factors that influence the susceptibility of some plant phenotypes can be identified [13]. For instance, transcriptomic analysis of the systemic defense response of *C. sinensis*, coinfected with mild and severe strains of CTV and ‘*Candidatus* Liberibacter asiaticus’ (CLas), indicated that some members of the WRKY gene family were upregulated. However, other members of this gene family were downregulated in the CLas coinfection case [14].

In several orchards in Veracruz, Mexico, trees showing symptomatology consistent with CTV infection were observed. These plants displayed diverse symptoms, suggesting the presence of different isolates. It was of special interest to analyze the host response at the transcriptome level in response to the naturally occurring CTV infections in orchards. This could provide information on the spatial dynamics of gene expression, and on the long-term response to viral infections in trees that were infected for at least several months. Our results are discussed in terms of the signaling, disease resistance and epigenetic responses to virus infection, as well as CTV–viroid coexistence.

## 2. Results

### 2.1. Detection of Different CTV Isolates in C. Sinensis Showing Mild and Severe Symptoms

The possible presence of CTV infections was observed in orchards of *C. sinensis* in the northern area of Veracruz, Mexico. Based on the presence of different symptoms in trees, six different locations were selected for sampling (Figure 1; Appendix A). Leaf samples were collected in several representative orchards. Closer inspection of symptomatic trees showed two different symptoms. Trees displaying only yellowing of fruits and leaves were classified as mildly symptomatic, while others showing dieback aside from the yellowing were considered as severely symptomatic (Figure 2). Trees with mild symptoms displayed general leaf vein yellowing (Figure 2A,B). In these cases, defoliation was also observed, but fruits were not affected (Figure 2C), and the stem showed a normal phenotype (Figure 2D). In general, trees with severe symptoms showed sparser foliage (Figure 2E), and leaf chlorosis was not only confined to the veins, but rather in all the leaf blade (Figure 2F). Fruits showed yellowing and smaller size compared to asymptomatic trees (Figure 2G). Notably, stem pitting was also observed in few trees (Figure 2H). The presence of CTV was determined by endpoint RT-PCR of the CP gene. From the 21 selected trees for collection, 12 were diagnosed as negative, and 9 trees were positive for the presence of CTV. Since sequence polymorphism linked to symptom severity among different CTV isolates has been reported [15], the CP ORF sequence was scanned for restriction site polymorphisms. Indeed, the severe isolate contained an EcoRI site in the position 500 of the CP ORF, which was absent in the mild isolate. Thus, enzymatic digestion of the RT-PCR products enabled to infer the type of isolate from the analyzed samples (Appendix A) and classified it as a mild isolate (MZ670756 or MZ670757) or a severe isolate (MZ670758). It is important to mention that control trees were classified as asymptomatic, rather than healthy, since two viroids were detected in these plants, as described below.

### 2.2. Phylogeny of Mild and Severe CTV Isolates

The phylogeny of both types of CTV isolates was determined based on the CP gene sequences and confirmed by the identification obtained with enzymatic digestion of the samples. MZ670756 and MZ670757 isolates were grouped in the clade that included mild strains such as the T30 isolate, which has been endemic in Florida for about a century [16]. T30 is a clade with low genetic diversity, considering the high similarity among isolates from Florida, Colima, Veracruz and Michoacán. On the other hand, the MZ670758 isolate was grouped with other similarly severe strains that cause stem-pitting symptoms, such as the CTV isolate from a declining sweet orange cv. Valencia tree (VT) from Israel [17] (Figure 3). In VT clade, three recognized groups were observed, isolated worldwide. A third, more divergent clade grouping severe isolates was represented in the upper branch of the tree (Figure 3). These strains are distributed in different citrus grown areas, including Asia, the known site of origin and diversity of citrus species. Interestingly, no isolates with homology to this clade were detected in the present study.

### 2.3. Bioinformatic Analysis of Naturally Infected Citrus sinensis Trees

The RNA-Seq data have been deposited in the Short Read Archive (SRA) of the National Center for Biotechnology Information (NCBI) with accession number PRJNA748945. A total of 64,807,664 pair-end raw sequences were obtained from the generated libraries for each condition: infected trees with mild CTV isolate (20,704,122 reads), infected trees with severe CTV isolate (20,994,616 reads), and asymptomatic trees (23,108,906 reads). A total of 63,381,946 high-quality reads were obtained as follows: infected trees with mild CTV isolate (20,262,774 reads), infected trees with severe CTV isolate (20,550,260 reads), and asymptomatic trees (22,568,912 reads) (Appendix A). A total of 23,953,005 paired sequences and 14,663,410 unpaired sequences were obtained. Before transcriptome assembly, longer sequences were obtained by overlapping regions (Appendix A). A total of 124,794 unigenes were generated. Finally, 83,672 corrected and non-redundant sweet orange representative sequences were obtained (Appendix A).

### 2.4. Identification of Differentially Expressed Genes

The differential accumulation of transcripts between asymptomatic and CTV-infected plants was analyzed: 1113 differentially expressed transcripts between asymptomatic and mildly symptomatic plants, and 1110 between asymptomatic and severely symptomatic plants were identified per a total of 2223 differentially expressed genes (Figure 4; Appendix A). The heat map results indicated that there was some overlap between the sets of significantly differentially accumulated transcripts (Figure 4A). The Venn diagram shows the differences and overlapping of accumulated transcripts in tested conditions (Figure 4B).

#### 2.4.1. Functional Classification by Gene Ontology (GO)

Gene Ontology terms were assigned from the non-redundant unigenes based on information of homologous species, in this case, *Arabidopsis thaliana*. For each functional cluster, the major 10 represented categories of the differentially expressed transcripts (DETs) are shown (Appendix A). These categories were mostly similar for both mildly and severely infected samples. Within the CC cluster, the most represented categories were nucleus, chloroplast, cytoplasm, other intracellular components, and cytosol, for the mild infection condition (Appendix A). In the case of the severe isolate, the most represented categories were nucleus, cytoplasm, chloroplast, cytosol, and other intracellular components (Appendix A). The most represented categories in the MF class for both conditions (trees infected with the mild and severe isolate) were catalytic activity, protein binding, transferase activity, hydrolase activity, and other binding (Appendix A). Finally, the most represented BP categories in plants infected with either the mild or the severe isolate were other cellular processes, other metabolic processes, response to stress, response to chemicals, and biosynthetic processes (Appendix A). The number of upregulated and downregulated genes was different for each category corresponding to mild or severe CTV-infected plants.

#### 2.4.2. Gene Enrichment Analysis by PlantGSEA and Functional Classification by KEGG

Major plant metabolic pathways affected by CTV infection with mild (Appendix A) or severe strains (Appendix A) were identified. Three salient routes were recognized: (1) defense response; (2) innate immune response; and (3) cell-to-cell transport (Figure 5). The higher significance levels of enrichment were found in the GO:0006952 gene set (defense response) (Figure 5A,B). These significance levels were higher in plants infected with a severe CTV strain than with a mild isolate. Similarly, the significance levels of enrichment were also higher in the innate immune response (GO:0045087) gen set of the severe-infected group (Figure 5C,D). In contrast, the gene sets related to cell-to-cell transport (cell junction (GO:0030054), cell-cell junction (GO:0005911) and plasmodesma (GO:0009506)) showed a higher level of coordination in trees infected with the mild strain (Figure 5E,F). It must be mentioned that the level of significance correlates with the number of genes of a given cellular pathway represented in a transcriptome set.

According to the DETs, several RNAs encoding potential disease-resistance proteins that contain a nucleotide-binding site and leucine-rich repeats (NBS-LRR) accumulated differentially in trees infected with either the mild or the severe CTV strain, with some overlap. In some cases, these were upregulated, while in others they were downregulated (Appendix A). Furthermore, transcripts for PR proteins PR1, PR4 and a PR5 isoform (thaumatin-like 1b) accumulated in high levels in trees infected with the mild strain (log_2_ fold change (FC): 5.8784, 4.9185, and 5.1851, respectively), while only PR5 was differentially downregulated in trees infected with the severe strain (log_2_ FC: −7.5925). Additionally, two basic chitinase mRNAs were induced in response to infection with the mild strain. In contrast, one of these was not induced and the other was downregulated in response to the severe strain (AT3G54420.1 (log_2_ FC in mild strain infection: 12.1969; in severe strain infection: −7.3038) and AT3G12500 (log_2_ FC in mild strain infection: 7.0524)). Thus, the general response to pathogen infection usually associated with systemic acquired resistance (i.e., PR proteins) was generally downregulated or not induced in response to infection with the CTV severe strain.

Interestingly, the light-harvesting chlorophyll-protein complex II subunit B1 mRNA was downregulated considerably in response to the mild and severe isolates (log_2_ FC: −8.5038), indicating that photosynthesis was affected similarly in both cases. In contrast, the photosystem I subunit G transcript was upregulated in the mild isolate infection (log_2_ FC: 8.4051). Interestingly, the photosystem II reaction center W mRNA was also upregulated in the severe isolate infection (log_2_ FC: 6.1085), but did not appear to be differentially expressed after mild isolate infection. Likewise, chlorophyll A-B-binding family protein mRNA and a probable chlorophyll(ide) b reductase NYC1 mRNA were similarly downregulated after infection with either isolate (log_2_ FC: −7.6366 and −8.5038). No other photosystem-associated protein mRNAs were found to be differentially expressed in both types of plants, and it is not clear what would be the net effect on photosynthesis after infection with either of these CTV isolates.

Positive single-stranded RNA viruses are absolutely dependent on the host’s translation machinery for their replication, as in the case of CTV. Hence, it was of interest to determine whether the expression of genes for ribosome assembly and function was affected by CTV infection. Several transcripts for ribosomal proteins were modulated by CTV infection. However, in most cases, these genes were downregulated. Furthermore, these gene sets showed little overlap between trees infected with the mild and severe strains. Thus, the translational machinery was negatively impacted by CTV infection, although different transcripts were affected upon infection with CTV mild and severe strains, respectively. Nevertheless, certain genes involved in DNA replication and repair, as well as in genome stability and integrity, were similarly affected, and most of these were downregulated (Appendix A).

The results obtained from the KEGG analysis showed the differences in the metabolic pathway in response to pathogens for both mild (Figure 6) and severe CTV strains (Figure 7). The Ca^+2^-dependent protein kinase (CDPK) and calmodulin (CaM)/calmodulin-like protein (CML) encoding transcript were overexpressed in both mild and severe CTV-infected plants. Whereas PR1, disease resistance protein RPS2 (resistance to *Pseudomonas syringae* protein 2) and dehydration-responsive proteinase (Rd19)-encoding transcripts were overexpressed only in mild CTV-infected plants.

Several genes involved in histone modification (mostly related to the repression of gene expression) were differentially expressed (Appendix A). Genes regulating chromatin remodeling as well as RNA processing were similarly differentially expressed. Interestingly, these genes were downregulated in trees infected with the mild or severe strain. For example, the AT4G02020.1 homolog (XP_006492341.1 protein ID in *C. sinensis*), which encodes a histone-lysine N-methyltransferase EZA1 isoform X2 was downregulated in the mild infection condition. Finally, a significantly higher number of transcripts per reads corresponding to the p20 and p23 CTV gene products that suppress the host PTGS response, were observed in the severe strain infection compared to the mild strain (Appendix A).

### 2.5. Viroid-Associated Sequence Reads

It was observed that both asymptomatic and severe CTV-infected trees contained a large number of reads corresponding to *Citrus exocortis viroid* (CEVd) and *Hop stunt viroid* (HSVd) during the sequence analysis of transcriptomes. For CEVd, the number of fragments per kilobases of contigs per million mapped reads (FPKM) was 4,528,301 in asymptomatic and 4,244,325 in CTV-severe isolates. For HSVd, 5,813,953 FPKM in asymptomatic and 5,639,097 FPKM in CTV severe isolates were found. Interestingly, no reads corresponding to either CEVd or HSVd were detected in mild CTV-infected trees (Table 1).

### 2.6. Validation of Differentially Accumulated Transcripts by RT-qPCR

Transcripts with the largest log_2_ FC were identified and selected in order to assess their differential accumulation by RT-qPCR. A total of 12 transcripts were selected: three upregulated and three downregulated for each evaluated condition (Table 2). Gene transcript accumulation levels compared to asymptomatic plants are shown (Figure 8). Results obtained for these sets of genes mostly agreed with the data from the transcriptome, validating differential gene expression.

## 3. Discussion

Emergent viruses threatening agronomically important cultivars have been described [18], and among them, those infecting citrus trees have been described in Mexico [19]. *Citrus tristeza virus* is one of the most important pathogens of citrus, and its high prevalence underscores the need to investigate the host response against infection by this virus. Another relevant consideration is the existence of several CTV strains, based on the elicited symptomatology, as well as the potential synergism or antagonism with the coinfecting pathogens. Several works have addressed the defense response induced by CTV in citrus cultivars. At the transcriptome level, the response of sweet orange to coinfection with mild and severe CTV strains, as well as both strains coinoculated with ‘*Candidatus* Liberibacter asiaticus’ (CLas) has been analyzed [13,14]. Upon experimental inoculation of two CTV strains, the severe strain CTV-B6 overtook the infection process in terms of symptomatology, and thus the mild strain CTV-B2 did not elicit cross protection against the former. Most affected genes were related to translation and photosynthesis [13]. Regarding coinfection of these strains with the causal agent of HLB, the mild strain exerted some antagonism with CLas. For instance, genes for the translational machinery were more negatively impacted in trees coinfected with the severe strain and CLas, compared to the mild strain and CLas [14].

In the present work, transcriptional response in naturally infected trees showing mild and severe symptoms corresponding to CTV infection was studied. A single nucleotide polymorphism was successfully used to identify mild and severe CTV strains. This represents a useful screening strategy to discriminate between mild and severe CTV isolates without the need to sequence the CP gene. For this study, phylogenetic reconstruction corroborated initial strain classification. End-point RT-PCR analysis of infected plants suggested that one strain was dominant over the other, although it cannot be discarded that the other strain was present at a much lower level.

The differences in transcript accumulation regarding the metabolic pathway in response to both mild and severe CTV strains showed interesting clues to understand the differences among mild and severe infections. For instance, CDPK and CaM/CML encoding transcripts were overexpressed in both mild and severe CTV-infected plants. CDPKs are major second messenger in plant signal transduction. They are mono-molecular Ca^(2+)^-sensor/protein kinase effector proteins, sensing Ca^(2+)^ signals transduced in protein phosphorylation cascades, described in abiotic and biotic stress signaling. Besides mediating stress-related and developmental processes, this route of signaling would contribute to PAMP-trigger immunity in a hypersensitive response [20]. PR1 was a highly overexpressed transcript of *Citrus* in mild infection, and its role for programmed cell death (PCD) has been established as the basis for plant–microbe interactions. Members of the PR1 family have been described to participate in resistance against pathogens. Interestingly, the PR1 family is represented in every plant species studied to date, and its homologues have been also found in animals, fungi, and insects [21]. Despite its importance in plant–pathogen interaction, no overexpression of PR1 was registered in severe viral infection in the analyzed plants. PR1 is involved in the suppression of cell death-dependent disease symptoms. PR1-interacting proteins, including some members of the Rac1 immune complex, are known to function in innate immunity in rice and animal systems [21]. The lack of capacity to induce the PR1 family could account for the severe symptoms observed in the plants. The unknown mechanism of severe CTV isolates’ ability to avoid PR1 overexpression is a fascinating model to understand this plant–virus interaction. RPS2 and Rd19 were overexpressed transcripts in mild CTV-infected plants. RPS2 contains leucine-rich repeat, membrane-spanning, leucine zipper, and P loop domains. It is associated with the bacterial secretion system pathway to elicit a hypersensitive response, and in defense signal transduction. It is involved in nucleotide triphosphate binding and protein–protein interactions, and likely associated with the reception of the pathogen proteins [22]. Rd19 is a cysteine proteinase involved in apoptosis and plant–pathogen interactions, and described to be induced by desiccation but not by abscisic acid [23,24]. It has been reported to be involved in plant developmental processes and stress [25]. Rd19 associates with the *Ralstonia solanacearum* type III effector PopP2 to form a nuclear complex that is required for the activation of the RRS1-R-mediated resistance response [26]. Its role in viral infections is yet to be described.

It must be emphasized that initial inoculation of the virus may have taken place 3–6 months prior to sample collection, and infection may have occurred several times. Thus, the host defense response may be continuously activated, which could lead to impaired growth. Another factor that could affect plant development is the downregulation of genes related to photosynthesis. In this case, this set of genes showed a similar expression pattern and levels of downregulation in response to mild and severe CTV infection. Primary general metabolism was likely to be affected, since mRNAs for certain enzymes in the glycolytic pathway (hexokinase and glucose-6-phosphate isomerase) were downregulated in both conditions, while chloroplast fructose-bisphosphate aldolase 3 was downregulated only in response to the mild CTV strain. In contrast, the glyceraldehyde-3-phosphate dehydrogenase (GAPC1) transcript was upregulated in both cases. Interestingly, this enzyme is translocated to the nucleus in response to heat stress, where it facilitates binding of the transcription factor NF-YC10 to heat-inducible gene promoters [27]. Conceivably, a similar response occurred due to the stress elicited by viral infection. Intriguingly, no differential accumulation was observed for mRNAs encoding enzymes of the tricarboxylic acid cycle. It was possible that the general metabolism in infected plants was adjusted after a period of time, at least in terms of gene expression, resulting in a sort of acclimation to the presence of the virus.

According to the differentially expressed gene analysis, the histone-lysine N-methyltransferase EZA1 isoform X2 transcript was included in most gene sets. This protein is a part of the chromatin-modifying machinery, which plays an essential role in the epigenetic regulation of chromatin. In turn, it has an important role in genome stability and the long-term regulation of gene expression [28]. Recent evidence indicates a role of epigenetic mechanisms modulating the defense response to pathogens [29]. Thus, we searched in the general transcriptomic data for deregulation of more transcripts coding proteins with related functions. Indeed, several genes involved in histone modification, chromatin remodeling regulation, and RNA processing were differentially expressed. The importance of the physical modification of chromatin and histones (changes in histone density and variants) has become evident, as regulators of defense responses to plant pathogens [30].

CTV harbors three genes that code for silencing suppressors: CP (p25), p20 and p23. The accumulation of transcripts of p20 and p23 was much higher in the severe strain than the mild strain, suggesting that the symptomatology might also result from an effective suppression of the RNA silencing defense machinery. Thus, engineering the post-transcriptional gene silencing (PTGS) pathway to counteract silencing suppressors may also help to mitigate CTV infection or other viruses that use a similar strategy. However, it was not clear whether the ability of the host to counteract the effect of silencing suppressors might be related to the induction of the suppressor of gene silencing (SGS3) mRNA, since this induction was only observed in trees infected with severe CTV strain. On the other hand, the downregulation of Argonaute-1 (AGO1) mRNA only occurred in response to severe CTV infection. Clearly, more studies are required to determine which host factors are able to counteract the viral silencing suppressors in citrus, which is related to the generation of siRNAs in response to virus infection [16,31]. CTV CP also acts as a silencing suppressor, and a single amino acid substitution affected symptom severity in *Nicotiana benthamiana* [32].

The presence of two viroids commonly coinfecting citrus trees (CEVd and HSVd) in asymptomatic plants suggested that the differential gene expression reported herein occurred in response only to CTV infection, and their presence was also found in severe CTV-infected plants. This was reinforced by the lack of induction of defense-related genes, such as PR proteins, in asymptomatic trees. The presence and co-existence of these viroids suggested that they may be widespread in orange trees, and perhaps in other citrus species. However, no reads corresponding to either viroid in mild CTV-infected trees were found. So, the co-existence of severe CTV strain and both viroids were observed, but not in the mild CTV-infected plants. Consequently, the possibility of synergism or antagonism between viroids and different CTV isolates emerged. It will be necessary to inoculate plants with a severe and mild CTV isolate and CEVd or HSVd to analyze such a possibility. Interestingly, the failure to induce genes encoding PR proteins might result in a more severe symptomatology, more so if there is synergism between the severe CTV strain and CEVd and HSVd. Gene editing resulting in a rapid and continuous induction of these genes via the activation of NPR1 may result in tolerance to CTV infection, or at least, a mitigation of symptoms. Indeed, it has been shown that the overexpression of NPR1 in citrus partially protected plants from CLas infection [27]. Interestingly, in a parallel study in cucurbits, the viral infection did not only affect mRNA differential expression, but also differences in transcript mobilization in the phloem translocation stream of pumpkin plants were observed [33].

The analysis of the host response to mild and severe CTV strains indicated that there was some overlapping in the expression patterns. In comparison with other studies, certain Arabidopsis homologs (e.g., AT5G51060 (respiratory burst oxidase homolog protein C, log_2_ FC: −9.7039 [34]), and AT4G02380 (senescence-associated gene, log_2_ FC: −6.7460 [35])), are differentially expressed in response to several types of stress and are likely part of a general response to such stimuli. Although in the present case, those genes that were downregulated were more abundant than those that were upregulated. On the other hand, in this work, while both strains affected the expression of genes involved in DNA repair, genome stability and translation, the genes were affected distinctively. Finally, the knowledge of the signaling pathways in response to CTV infection could help to design novel strategies for the control of CTV, such as the genome editing of relevant differentially expressed genes.

## 4. Materials and Methods

### 4.1. Plant Material Collection

Citrus leaf samples were collected in the northern area of the state of Veracruz, Mexico, in October 2017. Different samples were collected from a total of 21 trees: healthy (asymptomatic), showing mild symptoms consisting of yellowing, or showing severe symptoms (dieback, stem pitting, and yellowing). Additionally, leaf samples were collected from the 4 cardinal points, at an approximate height of 1.5 m. All samples were placed in 15 mL falcon tubes containing 10 mL of RNA stabilization solution (RNAstat; Biopure, Mexico). Samples were stored at 4 °C for further analysis.

### 4.2. RNA Extraction

RNA extraction from leaf tissue was carried out as described by Logemann [36]. The resulting pellet was resuspended in 30 µL RNase free water. RNA concentration and absorbance 260/280 ratio was determined with a Nanodrop One spectrophotometer (Thermo Scientific; Waltham, MA, USA). The RNA integrity was verified using denaturing agarose electrophoresis and visualization was carried out with a BioDocAnalyze gel documentation system (Biometra Gmbh, Jena, Germany).

### 4.3. CTV Molecular Detection

RT-PCR assays were performed using the RNA obtained from leaf samples as templates (diluted to 50 ng/µL) with a SuperScript III One-Step (Invitrogen, Carlsbad, CA, USA) commercial system, following the manufacturer’s instructions. Following primers were used to amplify a 688 bp fragment corresponding to the p25 (CP gene) [15]: R731-7 (5′-CGGAACGCAACAGATCAACG-3′) and RF-137 (5′-ATTATGGACGACGAAACAAA-3′). It was observed previously that severe CTV isolates contained an EcoRI site in the 5′ end of the CP gene. In order to determine the severity of the CTV strain, the RT-PCR products were digested with EcoRI and analyzed by gel electrophoresis. These products were cloned in pDrive (Qiagen, Hilden, Germany) and sequenced to confirm the presence or absence of this polymorphism.

### 4.4. Cloning, Sequencing and Phylogenetic Analysis of the CTV Isolates

The amplified products were cloned into the pDrive vector (Qiagen) following the manufacturer’s instructions. Resulting plasmids were purified using the alkaline lysis method and sent for sequencing at UBIPRO, FES Iztacala (UNAM, Mexico). Sequences were edited with the Seqtrace-0.9.0 software [37]. The sequences obtained were deposited in GenBank (Accession No.: mild isolate 1 (M1): MZ670756; mild isolate 2 (M2): MZ670757; severe isolate (S1): MZ670758).

CTV strain identification was performed based on phylogenetic analysis of the CP gene sequences (~672 bp). For the phylogenetic analysis, CP gene sequences from isolates previously identified on the basis on the severity of the elicited symptomatology [31,38] were obtained from NCBI (https://www.ncbi.nlm.nih.gov/, accessed on 14 February 2022). Multiple sequence alignment (MSA) was performed using the MAFFT software version 7 [39] (https://mafft.cbrc.jp/alignment/software/, accessed on 14 February 2022), and final sequence edition was performed manually for further gap elimination with MEGA X software [40]. The online version of IQ-TREE (http://iqtree.cibiv.univie.ac.at/, accessed on 14 February 2022) was used for the determination of the best partition schemes for the MSA and the best evolutionary model. The phylogenetic reconstruction was performed with the 10,000 ultrafast bootstrap iterations approach and Maximum Likelihood (ML) method with 100 trees. The ML ultrafast bootstrap tree was visualized with iTOL (https://itol.embl.de/, accessed on 14 February 2022). The CTV isolate was determined according to the clade grouping.

### 4.5. Library Preparation and Sequencing for Massive RNA Sequencing (RNA-Seq)

Leaves from 3 independent trees classified as plants infected with a mild CTV isolate, plants infected with a severe CTV isolate, and asymptomatic plants were pooled. Therefore, each sample consisted of three biological replicates. RNA was purified as described and treated with DNase I (Invitrogen, Carlsbad, CA, USA). Each sample was diluted to a final concentration of 0.13 µg/µL, and one volume (15 µL) of RNA stabilizing agent (RNAstat; Biopure, Mexico) was added. The samples were sent to Otogenetics Corporation (Atlanta, GA, USA) for sequencing. The RNA-Seq method used was based on poly (A+) selection, which enriches eukaryotic mRNA and other polyadenylated RNAs. Each selected sample was processed with the TruSeq RNA Sample Preparation Kit (Illumina, San Diego, CA, USA), and index codes were assigned to identify each sample independently. RNA sequencing libraries were generated using Illumina sequencing instrumentation (Illumina HiSeq2500). Multiplexed libraries were sequenced on a single-flow cell of the HiSeq2500 platform to generate 125 bp paired end reads. Each RNA-Seq set consisted of 20 million reads per sample.

#### 4.5.1. Bioinformatic Analysis

Data processing was carried out using the High-Performance Cluster at the Instituto de Ecología A.C. (INECOL; Xalapa Veracruz, Mexico). Quality of sequence reads was assessed using the FASTQC program. Prior to assembly, sequences were processed with a quality control script (https://github.com/Czh3/NGSTools/blob/master/qualityControl.py, accessed on 14 February 2022) using the parameters −q 20, −p 90, and −a 30 to obtain high quality paired sequences. SeqPrep v1.1 (https://github.com/jstjohn/SeqPrep, accessed on 14 February 2022) was used to merge overlapping paired reads with a minimum of 87% coincidence required in the overlapping region. Sequences that could not be merged were nonetheless used for transcriptome assembly of *C. sinensis* infected with CTV. High-quality merged and unmerged reads were de novo assembled with Trinity 2.4.0 [41] using default parameters. A single transcriptome was generated using high-quality reads from the 3 tested conditions (plant infected with a mild CTV isolate, a severe CTV isolate, and asymptomatic plant). Resulting contigs (unigenes) were processed with Deconseq [42] to get rid of contaminating sequences. AlignWise was used to identify open reading frames (ORFs) and their translated amino acid sequences. Insertions and deletions within and outside of ORFs were corrected [43]. Redundant sequences with more than 90% of identity between them were removed. Non-redundant sequences were obtained with BlastClust [44].

The resulting non-redundant translated sequences of the transcriptome were functionally annotated by assigning them the function of highest match resulting from the alignment with BLASTp (https://blast.ncbi.nlm.nih.gov/Blast.cgi?PAGE=Proteins, accessed on 14 February 2022) using an E-value threshold of 10^−5^. Three reference sequences (*C. sinensis*, *Citrus clementina* and *Arabidopsis thaliana*) were used as model genomes.

Global expression profiles were compared for each transcriptome set to identify differentially expressed genes. High-quality (HQ) sequences from each condition were mapped to the reference transcriptome (unigenes) with the RSEM package [45]. This program uses short sequence mapping tools such as Bowtie2 and yields the normalized expression profile as TPM (transcripts per million) and FPKM (fragments per kilobases of contigs/genes per million mapped reads). The resulting counts were processed with the R/Bioconductor DESeq package [46], which normalizes samples by pairwise comparison using algorithmic expression based on the hypothesis that most genes are not differentially expressed. A heat map was obtained with this data using the R software package. An adjusted *p* value < 0.05 was considered as the threshold to identify expressed genes. To compare differentially accumulated transcripts between tested conditions a Venn diagram was constructed using Venny software version 2.1 (https://bioinfogp.cnb.csic.es/tools/venny/index.html, accessed on 14 February 2022) [47].

#### 4.5.2. Gene Enrichment Analysis

Gene enrichment analysis was performed with the Plant GeneSet Enrichment Analysis Toolkit (PlantGSEA) online webserver (http://structuralbiology.cau.edu.cn/PlantGSEA/, accessed on 14 February 2022) [48]. The analysis was performed using the annotation of the CTV-infected *C. sinensis* (with severe and mild strain) transcriptome that was assigned from *A. thaliana*. For this analysis, the gene IDs corresponding to the following conditions were recovered: (1) trees infected with the severe CTV strain; (2) trees infected with the mild CTV strain. The list of genes for each condition was analyzed separately in PlantGSEA, selecting *A. thaliana* as a reference species. The analysis was carried out with the Gene Ontology Consortium server (http://geneontology.org/, accessed on 14 February 2022) at the following levels: biological process (BP), cellular component (CC), and molecular function (MF).

#### 4.5.3. Assignment of Specific Metabolic Pathways

The amino acid sequences of the differentially expressed genes were selected for each evaluated condition. The assignment of specific metabolic pathways was carried out using the Kyoto Encyclopedia of Genes and Genomes (KEGG) database. Automatic annotation and mapping service KEGG BlastKOALA (https://www.kegg.jp/blastkoala/, accessed on 14 February 2022) [23,49] was used for this analysis, using the taxonomy ID 2711 which corresponds to *C. sinensis.*

#### 4.5.4. Viroid-Associated Sequence Reads

CEVd and HSVd were identified by mapping the RNA high-quality reads from the transcriptome dataset using Bowtie2 software version 2.4.4 (http://bowtie-bio.sourceforge.net/bowtie2/index.shtml, accessed on 14 February 2022). Reference genome sequences for HSVd (Accession No. EU104682) and CEVd (Accession No. J02053.1) were used.

### 4.6. Validation of Differential Gene Expression with RT-qPCR

Upregulated and downregulated genes were selected and further analyzed with RT-qPCR to determine correlation with the RNA-Seq data, regarding their upregulation or downregulation in different conditions. Three independent trees corresponding to each condition were evaluated in triplicate. The reaction conditions were previously standardized by endpoint RT-PCR. The Kapa Sybr Fast One-Step (Roche, Boston, MA, USA) universal system was used following the manufacturer’s recommendations. Specific oligonucleotides were designed to target each gene (Appendix A). RNA was diluted to a final concentration of 50 ng/µL. Amplification was carried out in a Step One plus system (Applied Biosystems, Foster City, CA, USA) with the following set-up: 5 min at 42 °C, 3 min at 95 °C, and 40 cycles of 3 s at 95 °C, and 20 s at 60 °C. The equipment default program was used to obtain the melting curve. Relative expression levels were calculated with the 2−ΔΔCt method [50]. The mRNA of actin (ACT) was used as endogenous internal control to normalize gene expression in sweet oranges [51]. Values from three technical replicates per sample were averaged to calculate relative expression. Statistical analysis was performed using the Microsoft Excel Analysis ToolPak (Microsoft Corp., Redmond, WA, USA). Significant differences between the expression values of the endogenous internal control and CTV-infected samples were determined using Student’s *t*-tests.

## Figures and Tables

**Figure 1 ijms-23-02435-f001:**
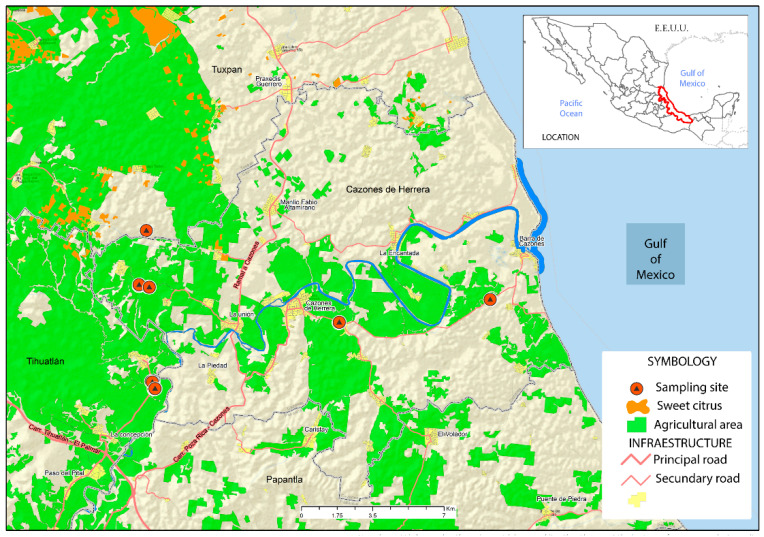
Location of collecting sites of sweet orange trees showing CTV symptoms in orchards in Veracruz, Mexico. Upper, right inset shows Mexico with the state of Veracruz labeled red. Green indicates the agricultural area. Red circles with a black triangle show the sites where 21 productive trees were sampled.

**Figure 2 ijms-23-02435-f002:**
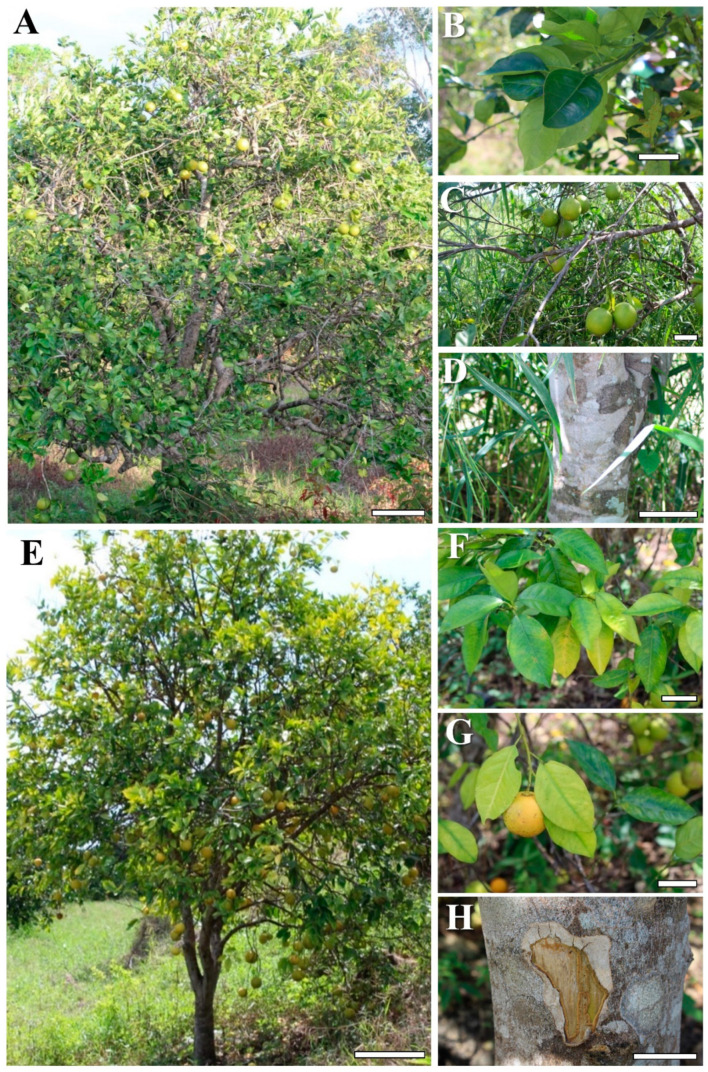
Contrasting symptoms of CTV-infected sweet orange trees in Veracruz, Mexico. *Citrus sinensis* trees infected with the mild (**A**–**D**), and the severe strain (**E**–**H**). (**A**) Sweet orange tree infected with the mild CTV strain; (**B**) leaves with yellowing of central vein; (**C**) rounded, normal orange fruits; (**D**) trunk with normal corky surface; € sweet orange tree infected with the severe CTV strain; (**F**) yellowing leaves; (**G**) orange fruits with yellow color and smaller size; (**H**) trunk with pitting damage. Bars in (**A**,**E**) = 50 cm; bars in (**B**–**D**) and (**F**–**H**) = 5 cm.

**Figure 3 ijms-23-02435-f003:**
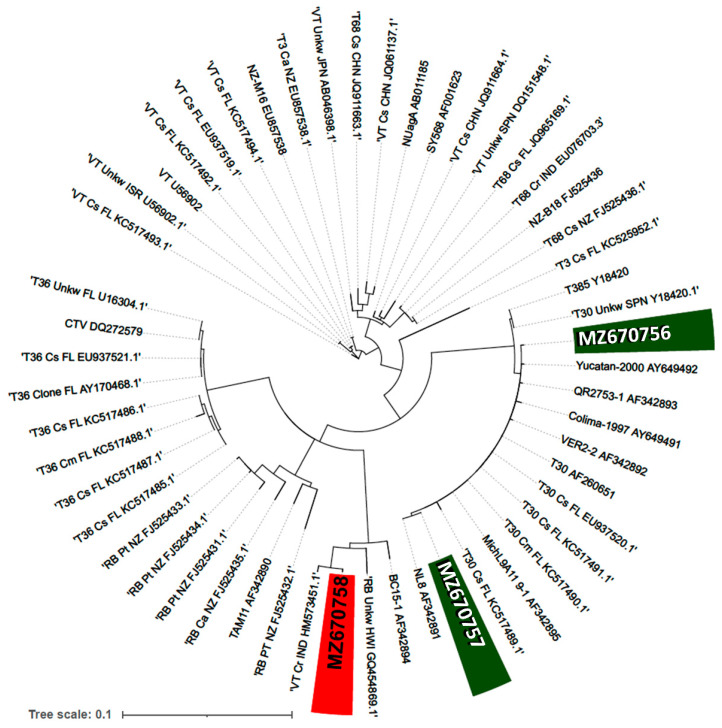
Phylogenetic relationships of CTV isolates based on the CP gene sequences. The Maximum Likelihood method was used for the phylogenetic tree inference. Upper semicircle shows a clade of severe CTV strains. Left semicircle shows a second clade of severe strains, including the strain identified in this study MZ670758. Right semicircle shows the clade with mild strains, including MZ670756 and MZ670757 identified in the present study.

**Figure 4 ijms-23-02435-f004:**
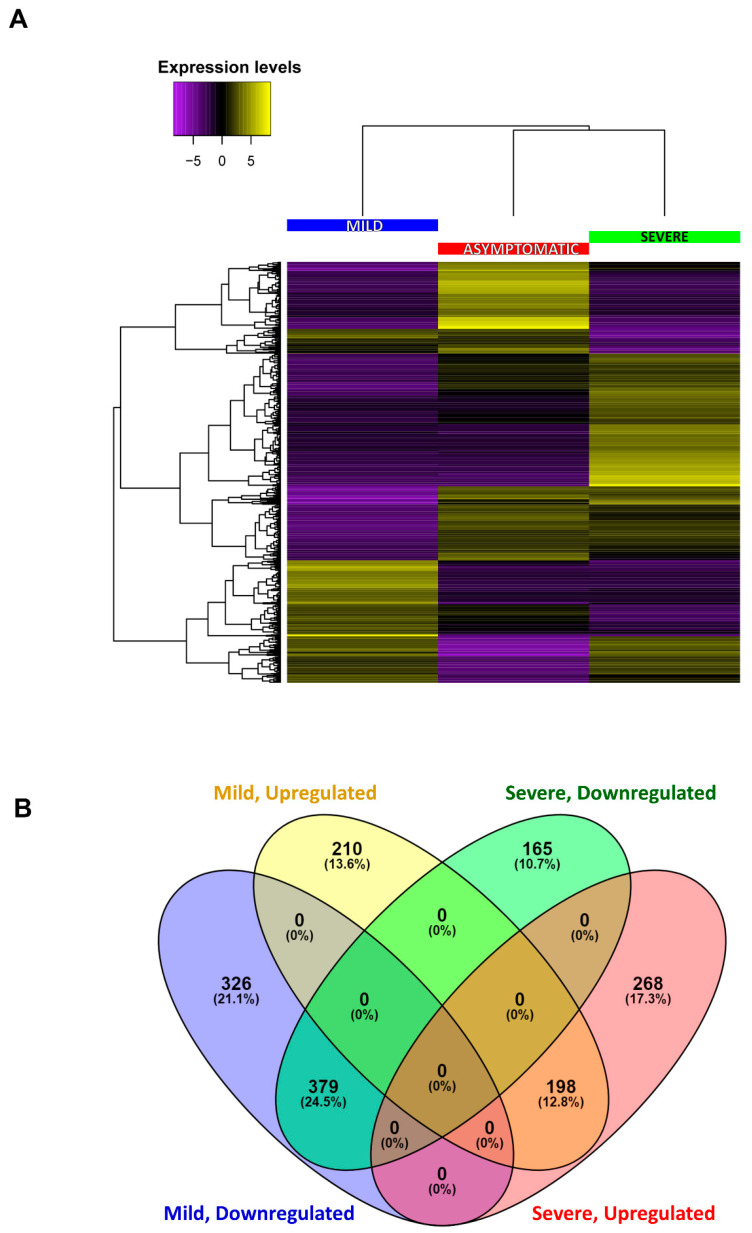
Gene expression levels in *C. sinensis* during CTV infection. (**A**) Heat map showing downregulated transcripts in violet, and upregulated transcripts in yellow. Transcripts from mildly infected plants (left column), asymptomatic plants (center) or severely infected plants (right) are shown. Genes with similar expression pattern are depicted in the left tree. (**B**) Venn diagram representing overlapped gene expression. Upper left: mild infection, upregulated transcripts; upper right: severe infection, downregulated transcripts; lower left: mild infection, downregulated transcripts; lower right: severe infection, upregulated transcripts. Numbers representing transcript hits and their proportion (percentage) are in parenthesis.

**Figure 5 ijms-23-02435-f005:**
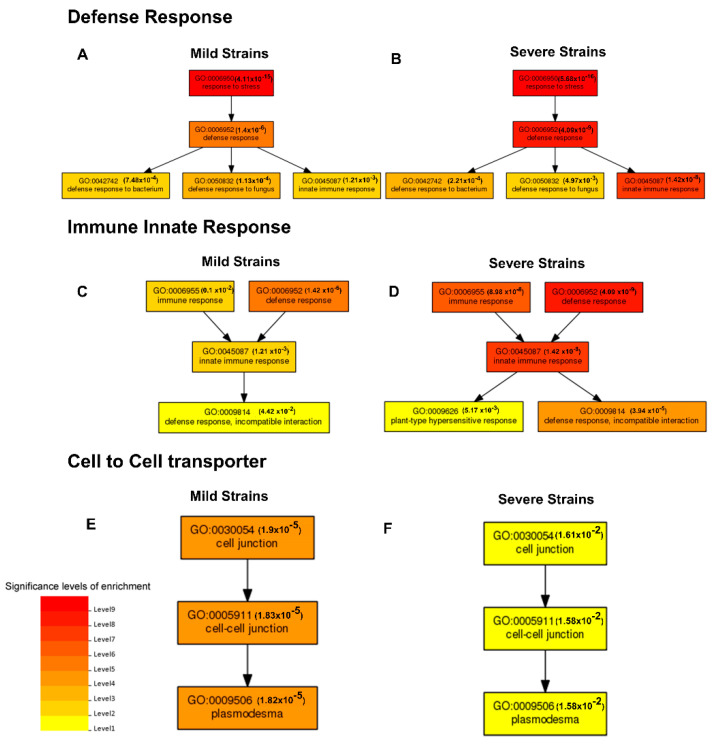
Comparison of differentially expressed transcripts in GO terms of biological functions. Left panels: responses to infection with mild strain; right panels: responses to infection with severe strain. (**A**,**B**) Defense response; (**C**,**D**) innate immune response; (**E**,**F**) cell to cell transport. Significance levels of enrichment are shown in red (upper levels) to yellow (lower levels).

**Figure 6 ijms-23-02435-f006:**
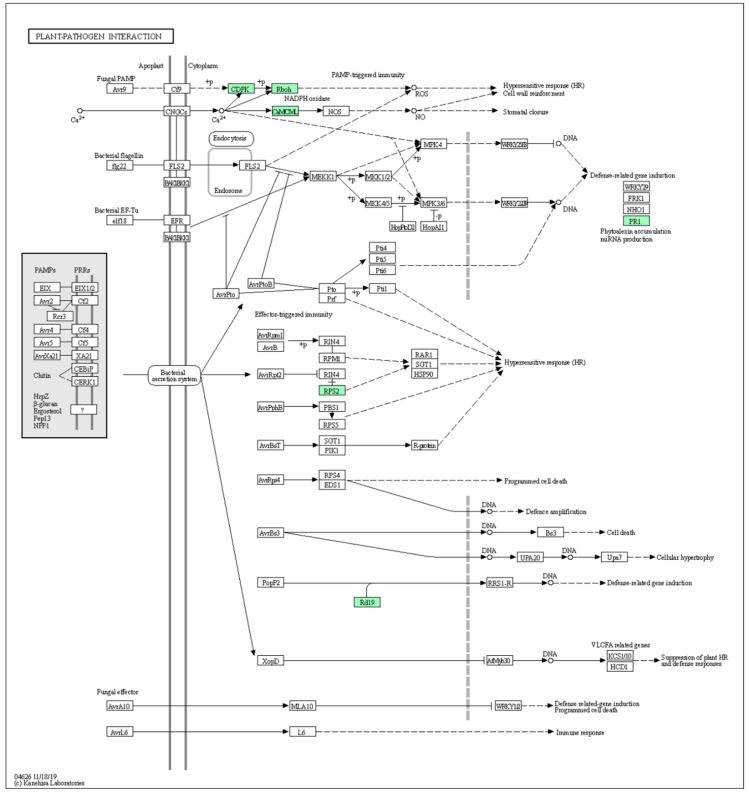
Upregulated transcripts associated with plant–pathogen interactions in the *C. sinensis* plant infected with the mild CTV strain. Green boxes indicate upregulated genes. Annotation was calculated with KEGG Blast Koala. Note the induction of CDPK, Rboh, CaM/CML PR1, RPS2 and Rd19.

**Figure 7 ijms-23-02435-f007:**
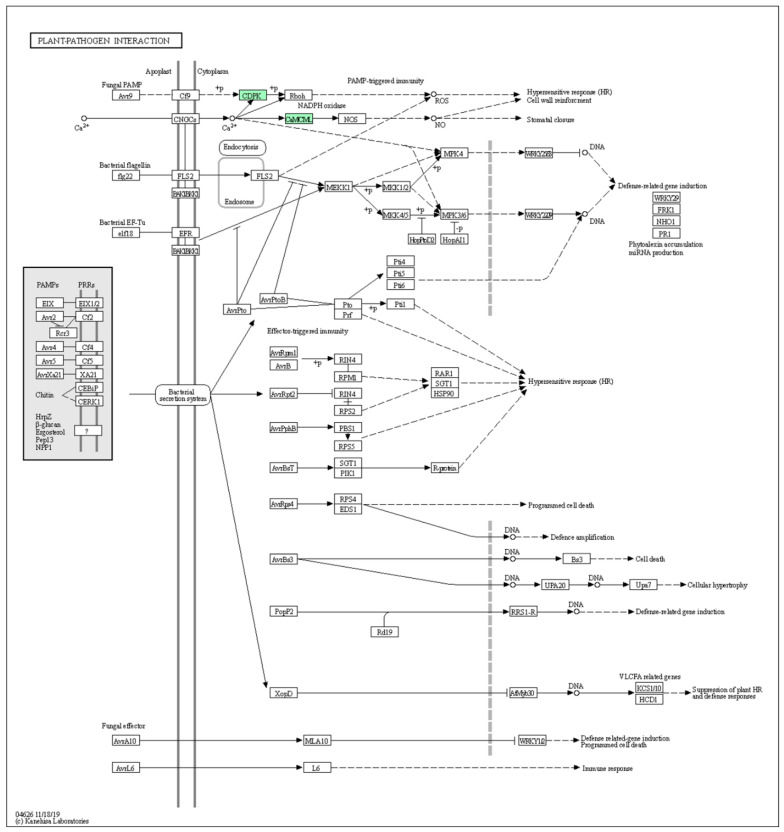
Upregulated transcripts associated with plant–pathogen interaction in the *C. sinensis* plant infected with the severe CTV strain. Green boxes indicate upregulated genes. Annotation was calculated with KEGG Blast Koala. Note the induction of CDPK and CaM/CML.

**Figure 8 ijms-23-02435-f008:**
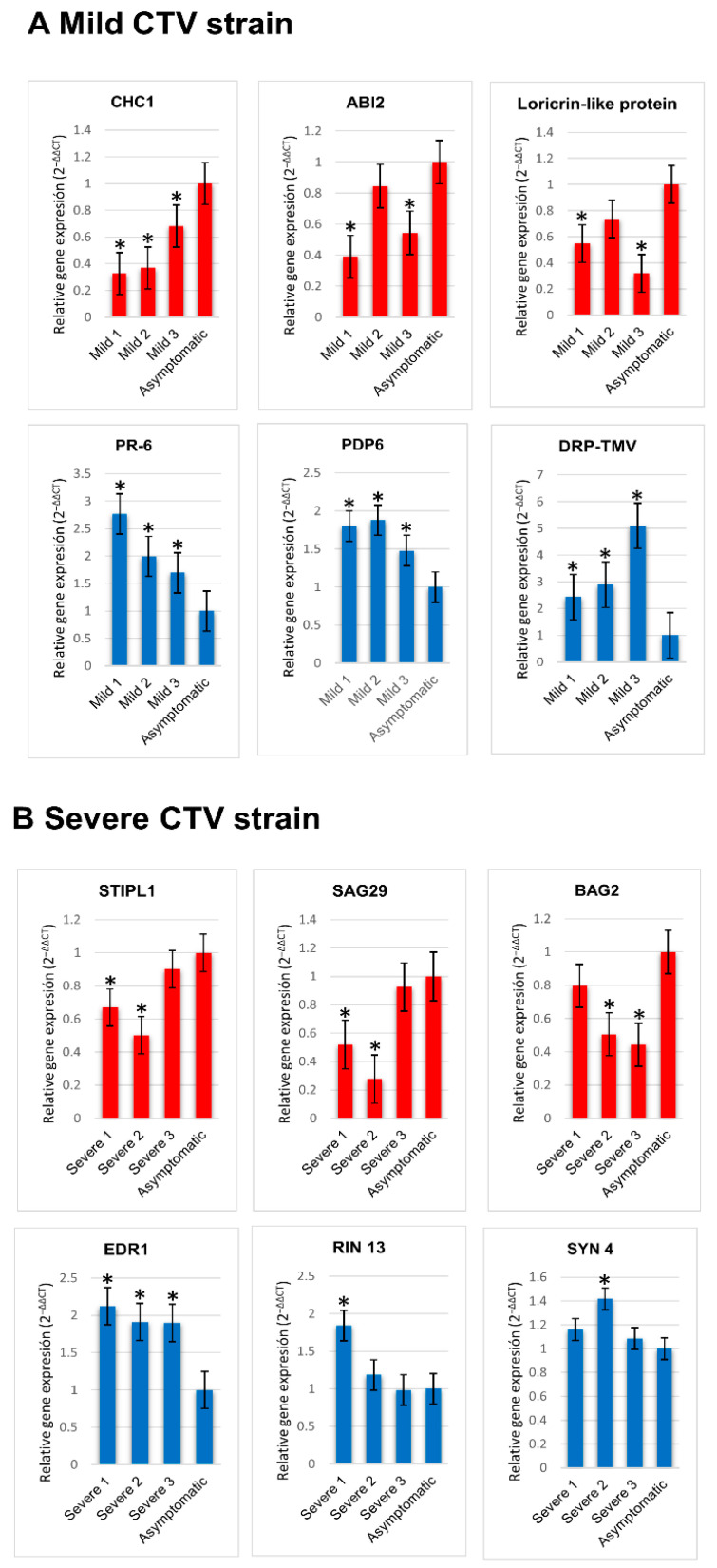
Validation of differential gene expression in CTV-infected trees. Expression levels of transcripts in *C. sinensis* infected with mild CTV strain (**A**) and severe CTV strain (**B**) were analyzed by quantitative RT-PCR. Detection was normalized with the endogenous transcript for actin. Transcript accumulation levels were compared to asymptomatic plants. Red: downregulated transcripts; blue: overexpressed transcripts. Mild CTV strain: downregulated genes (CHC1, ABI2, loricrin-like protein) and upregulated genes (PR-6, PDP6, DRP). Severe CTV strain: downregulated genes (STIPL1, SAG29, BAG2), and upregulated genes (EDR1, RIN13, SYN4). Three independent trees (Mild 1–3; Severe 1–3) were evaluated by triplicate. Bars represent standard deviation. Asterisks represent statistical significance (* *p* < 0.05).

**Table 1 ijms-23-02435-t001:** Number of reads of *Citrus exocortis viroid* and *Hop stunt viroid* in asymptomatic and CTV-infected orange trees.

Condition	TPM *	FPKM **
*Citrus exocortis viroid*		
Asymptomatic	1,000,000	4,528,301.89
CTV severe isolate	1,000,000	4,244,325.52
CTV mild isolate	0	0
*Hop stunt viroid*		
Asymptomatic	1,000,000	5,813,953.49
CTV severe isolate	1,000,000	5,639,097.74
CTV mild isolate	0	0

* The normalized expression profile is shown as TPM (transcripts per million). ** FPKM: fragments per kilobases of contigs per million mapped reads.

**Table 2 ijms-23-02435-t002:** Validation of differentially accumulated transcripts by RT-qPCR.

Gen ID	Gen Name	Description ^1^	log_2_ Fold Change
Downregulated genes, mild CTV strain			
XP_006470053.1	CHC1	CHC1 is predicted to encode a protein that belongs to the chromodomain remodeling complex.	−8.1749
XP_006475507.1	ABI2	Encodes a protein phosphatase 2C and is involved in ABA signal transduction.	−9.6421
XP_024949213.1	Loricrin-like protein	Loricrin-like protein; resistance to *Phytophthora.*	−7.3923
Upregulated genes, mild CTV strain			
XP_006490176.1	PR-6	PR (pathogenesis-related) peptide that belongs to the PR-6 proteinase inhibitor family.	5.0405
XP_006470311.1	PDP6	Histone H3-K27 trimethylation, regulation of transcription, DNA-templated.	7.9542
XP_024953872.1	DRP	Disease resistance protein (TIR-NBS-LRR class).	5.4683
Downregulated genes, severe CTV strain			
XP_006485913.1	STIPL1	Homologue of spliceosome disassembly factor NTR1. Required for correct expression and splicing of a regulator of seed dormancy.	−7.4838
XP_006483041.1	SAG29	Encodes a member of the SWEET sucrose efflux transporter family proteins.	−8.6582
XP_006468074.1	BAG2	Plant BAG proteins are multi-functional. They regulate apoptotic-like processes, pathogen attack to abiotic stress and plant development.	−7.2095
Upregulated genes, severe CTV strain			
XP_006471773.1	EDR1	Enhanced disease resistance 1 (EDR1) confers resistance to powdery mildew disease caused by the fungus *Erysiphe cichoracearum.*	7.1396
XP_006479423.1	RIN13	Encodes RPM1 Interacting Protein 13 (RIN13), a resistance protein interactor shown to positively enhance resistance function of RPM1.	7.2574
XP_024949989.1	SYN4	Encodes a SCC1/REC8 ortholog that may be involved in mitosis and may represent a mitotic cohesin. Plays a role in somatic DNA double-strand break damage repair.	7.0768

^1^ Source: https://www.arabidopsis.org/, https://www.ncbi.nlm.nih.gov/, accessed on 14 February 2022.

## Data Availability

The RNA-Seq data have been deposited in the Short Read Archive (SRA) of the National Center for Biotechnology Information (NCBI) with Accession No. PRJNA748945.

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
