# Peer review of "Transcriptomic Analysis of the Host Response to Mild and Severe CTV Strains in Naturally Infected Citrus sinensis Orchards"

_ijms, 2022, doi:10.3390/ijms23052435_

Round 1

Reviewer 1 Report

In the present study, authors have performed the RNA-Seq experiments to understand the responses of Citrus plants naturally infected by Citrus tristeza virus. They have included asymptomatic, mild and severely infected plant samples for their study. The major drawbacks observed in the study are as follows:

Authors have tried to make good sense of the transcriptomic data but somehow failed to represent the results clearly. The accession numbers provided for the nucleotide sequences and for the RNA-Seq data (PRJNA748945) could not be verified in NCBI. The table and figure numbers could not be matched appropriately as per the text description and it is really confusing for the reader to make good sense of it.

Moreover, sufficient experimental evidence suggesting the role of RNA silencing suppressors p20 and p23 genes was not provided. qRT-PCR validation was not done for any of the genes that were considered as the potential candidates by the authors.

Additional comments/suggestions:

  1. Introduction should include literature about the transcriptomic data on Citrus and also on Citrus Exocortis Viroid (CEVd) and Hop Stunt Viroid (HSVd). I felt that there is no need for detailed discussion about the results in the last paragraph of the introduction.
  2. Line 34: Correct as “biotic and abiotic factors”
  3. Line 46: correct as “do not affect the yield substantially”
  4. Line 54: Italicize the organism names wherever applicable for example in line 54- “C. aurantium” and line 190 “Arabidopsis thaliana”
  5. Line 69: Spell check “non-expressor”
  6. Line 70: Reference to be written normally
  7. For figure 2, please incorporate better figures comparing the leaves and fruits of control, mild and severely infected plant
  8. Although 9 samples were tested positive for CTV why only 3 isolates were considered is not clear. Please describe
  9. Fig S1: mark the ladder fragment sizes or mention the ladder used
  10. Line 139: Elaborate “VT”
  11. Figure 3: Mention right semicircle in the figure legends on line 149
  12. Rephrase it as “A total of 64,807,664 pair-end raw sequences
  13. It was mentioned that for RNA-Seq analysis three biological replicates were used but the Table S3 shows data for only 2 replicates. Please correct
  14. Fig 4A, change the font color to improve the visibility
  15. Supplementary file 1 mentioned in line 176 is missing from the supplementary file. Please correct
  16. Section 2.4.1: GO terms were mostly similar for both mild and severely infected samples. So, it would be good to either rewrite the section describing the percentage of genes that were differently expressed in mild and severely infected samples or the section can be merged with the previous section 2.4
  17. High resolution or better-quality images should be provided with better font color and size to improve visibility
  18. Is Table S4 mislabeled as S7? The description in results section doesn’t match the contents of the table provided so please correct
  19. Table S4, S5 & S6 are missing in the supplementary files. Please include
  20. Please provide the RNA-Seq expression raw data for all the gene mentioned in the present study as a supplementary excel file.
  21. For qRT-PCR experiment, it would be good to take the average of 3 biological replicates and represent the data for all the down and upregulated genes together with statistical significance.
  22. Figure 6 legend: What was the housekeeping gene used for the qRT-PCR? In Figure 6 legend, it is described as cytochrome c-oxidase but in the text, it is described as actin. Please check and include the details correctly. There is no need to give the description of genes studied in the figure legend.
  23. For fig 6., please mention what are the values on Y-axis.
  24. In addition of the genes validated through qRT-PCR, it would be really good to include some of the PR genes for validation in order to have more confidence of the observations made in the study.
  25. The paragraph discussing about the upregulated gene identified through KEGG analysis after section 2.7 should be merged with Section 2.4.2 as it would be more appropriate.
  26. Line 359, CaMCML was upregulated in both mild and severe isolate so please correct the sentence similarly in line 471, it was mentioned that AGO1 was downregulated only in severely infected plants but as per the table 1, it is downregulated in both mild and severely infected plants so please check again to incorporate the results accurately.
  27. The accession numbers of the isolates mentioned in the manuscript could not be found in NCBI GenBank
  28. Phylogenetic analysis was done using gene sequences or amino acid sequences? It is mentioned differently in the figure legend and in the methods section. Please check and verify
  29. Methods section does not describe the protocol for detection of viroids in the selected samples.

Reviewer 2 Report

Please find the attachemnt with detailed correction of your manuscript

peer-review-17165133.v1.pdf The manuscript is interesting but it still requires a lot of work. English must be definitely corrected by a native speaker. Many sentences are too long or they are difficult to understand, s they should be re-written. Introduction should be much longer and there should be described state of knowledge concerning a role of citrus in agriculture and its disease caused by viruses especially CTV. Please improve it. Results Results are not described appropriately. Authors included in this sections many information which should be  rather included in Materials and Methods or Discussion. This paragraph has to be re-written and complemented. Discussion also has to be corrected and a few more current references should be included.

Author Response

Thank you for the recommendations and suggestions. All your comments have been incorporated into the manuscript. Please find the attachment with the response for each comment.

https://drive.google.com/file/d/1ABU-Ahp9_rd0Vw02qksHTMZ9H6C-CgJK/view?usp=sharing

Round 2

Reviewer 1 Report

I really appreciate the authors for considering all the suggestions and incorporating necessary changes to the manuscript. The manuscript can be accepted with the incorporation of few minor corrections as suggested below:

  1. Fig 2. is repeated twice. Please edit
  2. Figures 6 and 8 need better resolution.

Author Response

  1. The figure 2 is repeated twice

This was corrected.

  1. Figures 6 and 8 need better resolution

Both figures have 300 dpi now.

Reviewer 2 Report

Please find the attachment of the manuscript with only few minor errors that reqiured correction. The Authors did an excelent work. The manuscript is much improved and should be accepted for publication in IJMS.

Author Response

  1. The original sentence: This contributed to provide information to better understand host-pathogen interaction.

Was rephrased to:

This work may contribute to our understanding of Citrus-virus interaction in natural settings, which can help develop strategies for integrated crop management.

  1. (CIVT, 2022) was removed.
  2. “Citrus” was changed to “citrus”